# MAGENTIC MARKETPLACE: AN OPEN-SOURCE ENVIRONMENT FOR STUDYING AGENTIC MARKETS

## ABSTRACT

As LLM agents advance, they are increasingly mediating economic decisions, ranging from product discovery to transactions, on behalf of users. Such applications promise benefits but also raise many questions about agent accountability and value for users. Addressing these questions requires understanding how agents behave in realistic market conditions. However, previous research has largely evaluated agents in constrained settings, such as single-task marketplaces (e.g., negotiation) or structured two-agent interactions. Real-world markets are fundamentally different: they require agents to handle diverse economic activities and coordinate within large, dynamic ecosystems where *multiple* agents with opaque behaviors may engage in open-ended dialogues. To bridge this gap, we investigate *two-sided agentic marketplaces* where Assistant agents represent consumers and Service agents represent competing businesses. To study these interactions safely, we develop *Magentic Marketplace*– a simulated environment where Assistants and Services can operate. This environment enables us to study key market dynamics: the utility agents achieve, behavioral biases, vulnerability to manipulation, and how search mechanisms shape market outcomes. Our experiments show that frontier models can approach optimal welfare—but only under ideal search conditions. Performance degrades sharply with scale, and all models exhibit severe first-proposal bias, creating 10-30x advantages for response speed over quality. These findings reveal how behaviors emerge across market conditions, informing the design of fair and efficient agentic marketplaces.

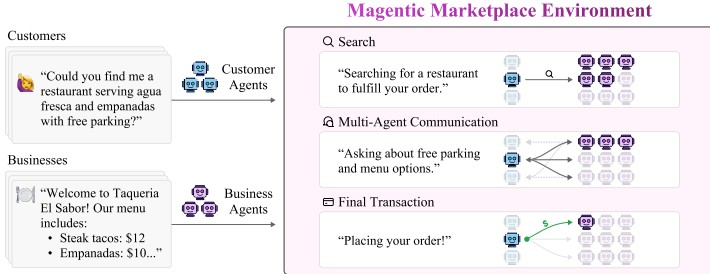

Figure 1: *Magentic Marketplace* is an open-source environment where AI agents can discover, communicate, and transact with each other. The environment can be used for evaluating different market designs and agent behaviors.

## 1 INTRODUCTION

Autonomous agents powered by large language models (LLMs) demonstrate rapidly expanding capabilities, ranging from software development and customer service to strategic negotiation and complex decision-making (Dong et al., 2025; Robeyns et al., 2025; Li et al., 2024a; Cui et al., 2017; Huang et al., 2025; Eigner & Händler, 2024; Hua et al., 2024; Abdelnabi et al., 2024a; Ferrag et al., 2025). As these capabilities mature, they create the foundation for multi-agent ecosystems where users can delegate economic activities to AI proxies that autonomously search and transact on their behalf (Hao & Xie, 2025; Karten et al., 2025b; Liu et al., 2024). The proliferation of such agents

in markets is poised to have a disruptive impact on economic activity, creating an urgent need for a deeper understanding of multi-agent economic behavior.

**Background: Agentic Markets.** Major economic platforms like Amazon, Facebook Marketplace, Google, and Bing are *two-sided markets* where consumers (on one side) and businesses (on the other) discover and transact with each other. Autonomous agents have appeared on both sides of these markets, such as shopping agents that mimic a human consumer navigating websites (*e.g.*, OpenAI Operator) and customer support agents that assist businesses in answering consumer queries (*e.g.*, Amazon Rufus and Expedia Romie). At present, these agents are designed to act as proxies for humans on one side of the market, with the implicit assumption that the other side is non-agentic.

Rothschild et al. (2025) argues that dramatic shifts will occur when *both* sides are simultaneously represented by agents that *interact with each other* in a *two-sided agentic market*. Two-sided agentic markets promise to generate added value by reducing communication costs and information asymmetries (*e.g.*, a business may not list every product configuration on its website). Humans cannot discover bespoke configurations without costly communication (*e.g.*, phone calls). Shopping agents that mimic human website-browsing encounter a similar information asymmetry. Agent-to-agent interaction, however, can overcome such an asymmetry by inexpensively engaging in conversation to explore the full range of possible options, generating value for both consumers and businesses.

There are many design decisions needed to architect and operationalize two-sided agentic market-places and many open questions about how current SOTA LLMs would perform under different market implementations. One critical challenge is to develop protocols that extend legacy designs for human consumers and businesses to allow for friction-less agent-to-agent interactions while allowing for human-human market interactions. At the same time, there is a growing commercial interest in implementing two-sided agentic markets, with companies like Google launching agent-to-agent communication and payment protocols (A2A, AP2). Current research on agentic systems has focused mainly on individual agent performance and structured interactions between agents for isolated economic tasks (Wang et al., 2025; Buscemi et al., 2025; Mao et al., 2024; Abdelnabi et al., 2024b). Our work goes beyond existing studies to capture the complex dynamics of two-sided agentic markets for future design decisions end-to-end.

**An Open-Source Environment.** We propose an agent-marketplace research paradigm, centering on the use of simulation environments for empirical studies of the capabilities and risks of LLM-based agents in multi-agent economic ecosystems. In particular, we introduce *Magentic Marketplace*, a simulated *multi-agentic marketplace* environment for controlled experimentation in agentic markets. The environment supports the full transaction lifecycle: from search and matching to negotiation and transaction, enabling systematic study of agent behavior under realistic marketplace conditions (see Figure 1 for an example). This simulation environment enables one to investigate questions such as: How effectively can agents discover and transact with one another? How do market design decisions impact agent efficacy at scale? How does current AI agent technology compare to ideal agentic behavior and non-agentic markets? How do agents behave in response to strategic and competitive market environments, relative to classic economic predictions?

Using *Magentic Marketplace*, we implement an experimental market scenario where agents seek to optimize outcomes for the consumers they represent, maximizing individual utility and generating rich interaction data. To enable controlled, repeatable experiments and safe exploration of agent behaviors, our current study uses fully synthetic data *e.g.*, from a restaurant domain (specifically, Mexican restaurants and contractors). But the environment is extensible: it supports additional synthetic domains and public/open datasets, facilitating research that shows generalization across market settings. We demonstrate how this setup can be used to evaluate market efficiencies under search limitations, susceptibility to manipulation. Our results reveal systematic behavioral biases and vulnerabilities across models, underscoring the need for further advancement of both agents and market mechanisms. Together, this paper establishes an empirical foundation for understanding the capabilities and risks of LLM-based agents in multi-agent economic ecosystems. Our contributions are as follows:

1. We design and implement the *Magentic Marketplace* environment to study LLM agents end-to-end across the two-sided economic market lifecycle, including search, inquiry and potential negotiation, and transactions.

2. We instantiate *Magentic Marketplace* with synthetic consumer and business data to measure economic welfare gains from two-sided agentic markets and understand their performance and vulnerabilities to manipulation and bias.

3. We open-source *Magentic Marketplace* to help others build multi-agent market designs, test new agentic solutions in these settings, and contribute experiment protocols to explore additional marketplace behaviors of LLM-agents.

## 2 RELATED WORK

The study of agents in marketplaces predates LLMs, with early work focusing on algorithmic agents, their interactions with human counterparts, and the implications for market outcomes (Wellman et al., 2004; Shahaf & Horvitz, 2010). More recently, the growing potential for an agentic economy has motivated the study of markets populated by AI agents. Researchers are investigating the forces that create and shape such markets as well as the conceptual benefits and risks of different designs (Hammond et al., 2025; Rothschild et al., 2025; Tomasev et al., 2025; Hadfield & Koh, 2025). Building on this foundation, *Magentic Marketplace* enables controlled experimentation of agentic economies.

**Economic Agents.** In agentic markets, AI agents are involved in making economic decisions. Prior work investigates the strategic and reasoning capabilities of agents in business and consumer decision problems (Allouah et al., 2025; Brand et al., 2023; Anthropic; Horton, 2023; Hua et al., 2024; Raman et al., 2024), offers/negotiation (Aher et al., 2023; Lewis et al., 2017; He et al., 2018; Liu et al., 2025; Godfrey et al., 2025; Zhou et al., 2025; Zhu et al., 2025), and bidding (Chen et al., 2024). These results indicate AI agents might be able to navigate markets on behalf of humans, something we test in *Magentic Marketplace*. There is a growing body of empirical studies of multi-agent economic interactions. Two-agent studies provide crucial insights, demonstrating collusion and the impact of personality, persuasion and other behavioral tactics on outcomes (Fish et al., 2025; Huang & Hadfi, 2024; Shapira et al., 2025a). Many-agent studies explore diverse scenarios such as optimizing tax policies (Zheng et al., 2020), group-think behaviors in competitive settings (Raghavan, 2025), optimal matching between people (Liang, 2025), and macroeconomic simulations (Li et al., 2024b). While these studies provide valuable insights, they examine isolated scenarios and abstract games. *Magentic Marketplace* enables many-agent interactions in marketplaces and exposes quantifiable business metrics, allowing researchers to systematically evaluate experimental performance and vulnerabilities.

**Economic Environments.** Other works provide frameworks and environments to simulate and study agents in economic games. These include benchmarks and evaluation suites that test economic rationality of LLMs and AI agents (Shapira et al., 2025b; Guo et al., 2024; Horton, 2023; Raman et al., 2024) as well as platforms for agent-led economic behavior such as the AgentExchange for task auctions (Yang et al., 2025c). Broader economic simulation work includes agent-based financial market modeling (Dwarakanath et al., 2025; Karten et al., 2025a), negotiation platforms (Bianchi et al., 2024), and social behavior simulation (Park et al., 2023). *Magentic Marketplace* goes beyond isolated games or behavioral studies by providing an end-to-end environment for enabling systematic study of persistent, many-to-many customer-business relationships across complete transaction lifecycles, from search and discovery through negotiation to fulfillment. Recent benchmarks have also explored asynchronous agent interactions (Andrews et al., 2025) and long-term coherence in economic decision-making (Backlund & Petersson, 2025), though these focus on general agent capabilities or isolated business scenarios rather than two-sided marketplace dynamics.

## 3 MAGENTIC MARKETPLACE

In this section, we first establish design goals for the environment that we built for studying agentic markets and then overview its implementation. Then we zoom in on the marketplace *protocol* that allows agents to register and discover capabilities, and finally detail how this protocol exposes specific actions that enable agents to execute the complete economic lifecycle from discovery to transaction.

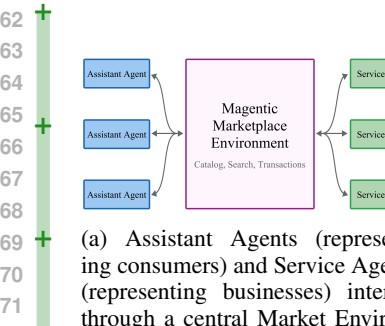 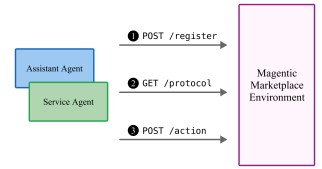 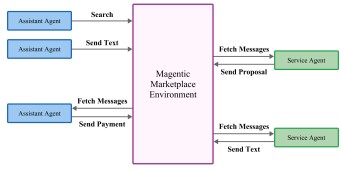

(a) Assistant Agents (representing consumers) and Service Agents (representing businesses) interact through a central Market Environment.

(b) Our implementation comprises three core endpoints: **register**, **protocol**, and **action**.

(c) Our protocol consists of 5 unique actions: search, text messages, order proposals, payments, and fetching messages.

Figure 2: Overview of *Magentic Marketplace*'s architecture: agents, endpoints, and action space.

## 3.1 ENVIRONMENT DESIGN GOALS

**Modeling Two-Sided Agentic Markets (Figure 2a).** Our environment simulates two-sided agentic marketplaces: platforms that connect **agents** acting with decision-making authority on behalf of human **principals** on both sides of a market. There should be two types of agents and their respective principals: **Assistant Agents** should act on behalf of customers and interpret **user intentions** to satisfy them, *i.e.*, engaging in dialogue with users to clarify needs and preferences, searching for suitable services, negotiating terms with Service Agents, and executing confirmed transactions. **Service Agents** should act on behalf of businesses and maintain internal catalogs of services provided, such as **POS** (Point of Sale) systems that track inventory, manage pricing, and process orders.

*Magentic Marketplace* should be general enough to capture the unique opportunities and challenges of such two-sided agentic markets. This includes agents with decision-making authority (*e.g.*, choosing whom to transact with) that navigate asymmetric and private information about potential transactions. Specifically, Assistant Agents don't initially know which businesses can fulfill requests or at what price, while Service Agents don't know customers' budgets or preferences: agents discover matches through conversational exchanges. This also includes indirect network effects: when multiple businesses offer similar services, competition on price and quality benefits consumers, and vice-versa. Finally, the design must prevent closed "walled gardens" by ensuring agents can freely discover and communicate with any other agent in the marketplace (Rochet & Tirole, 2003; Rothschild et al., 2025).

**End-to-End Economic Lifecycle.** We design for complete end-to-end economic processes with the goal of supporting the full transaction lifecycle from search and discovery through negotiation to fulfillment. This comprehensive approach enables systematic study of agent behavior under realistic marketplace conditions, capturing the complex emergent dynamics that characterize real-world economic interactions. The **environment** should supply all necessary infrastructure and manage market-wide capabilities including maintaining **catalogs** of available services, implementing discovery algorithms, and facilitating agent-to-agent **communication** including inquiry and negotiation. The environment should provide a centralized **transaction** layer that handles monetary exchanges and maintains transaction integrity across all marketplace interactions.

**Experimental Control.** The environment should enable systematic research across diverse agent implementations and evolving marketplace capabilities. Researchers should be able to: (a) integrate different agent architectures (LLM-based, rule-based, hybrid) in controlled studies, (b) evolve marketplace capabilities over time (adding refunds, reviews, ratings, etc.) without breaking existing experiments, and (c) ensure findings generalize to real-world deployment scenarios (MCP integrations) while maintaining precise control over experimental variables.

**Potential Research Directions** This environment enables a wide range of research directions at the intersection of AI agents, market design, and human-computer interaction. For example: How can we develop optimal Assistant Agents and Service Agents? What indexing and search mechanisms enable efficient agent discovery in large-scale marketplaces with heterogeneous services? How to design an efficient and effective communication protocol between agents? What mechanisms ensure truthful representation, prevent manipulative practices, and maintain transaction security in autonomous agent interactions? How can we design interfaces and interaction patterns that allow

humans to effectively supervise, guide, and override agent decisions when needed? How to design efficient serving system for such large-scale marketplace? By spanning challenges from agent design and information retrieval to human-AI interaction and distributed systems, *Magentic Marketplace* provides a comprehensive testbed for agentic market research. While the environment can be used to study many research questions, we narrow down the specific ones we'll explore in Section 4.

## 3.2 IMPLEMENTATION OVERVIEW

To achieve these design goals: two-sided marketplace structure, end-to-end economic lifecycle, and experimental control, we make three important architectural choices:

**1. HTTP/REST Client-Server Architecture:** Agents operate as independent clients while the marketplace environment serves as the central server, communicating through HTTP/REST endpoints. This enables **two-sided marketplace structure** through clear separation of customer and business agent roles. For **real-world applicability**, this design mirrors existing commercial platforms (Shopify, Amazon, eBay) and emerging agent protocol standards (MCP, A2A), allowing integration with existing infrastructure. The action-observation loop provides the foundation for studying marketplace behaviors while maintaining experimental control.

**2. Minimal Three-Endpoint Market Protocol (Figure 2b):** Supporting the **end-to-end economic lifecycle** requires many functionalities (search, communicate, negotiate, pay), but many endpoints hinder **experimental control**. We address this tension by designing three endpoints—register, protocol discovery, and action execution—that push complexity into the action space. Agents discover available actions dynamically, allowing new capabilities without breaking existing agents.

**3. Rich Action Protocol (Figure 2c):** Within the action endpoint, we design message types enabling the **two-sided marketplace structure** to support the complete **end-to-end economic lifecycle**: search (discovery), communication (negotiation), order proposals (structured offers), and payments (transaction completion). API specifications are in Table 1.

**Agent Action Protocol:** As shown in Figure 2c, the architecture enables the multi-phase agent lifecycle through five core actions: **Search** returns service agent lists, **Send Text Messages** facilitates communication, **Send Order Proposals** structures offers with items and prices, **Send Payments** accepts proposals, and **Receive** handles asynchronous responses. Assistant Agents initiate discovery and transactions (customer-driven), while Service Agents respond with messages and proposals (business-responsive). Both can Receive messages, enabling bidirectional negotiation.

## 4 EXPERIMENT DESIGN

Agents based on frontier models could improve market efficiency, but the fact that they are trained on human digital footprints raises concerns about inherited biases and vulnerabilities. In addition to these risks, such models introduce new potential vulnerabilities–such as susceptibility to prompt injection attacks and other forms of manipulation. These issues warrant systematic study. Accordingly, we use *Magentic Marketplace* to understand:

1. **Impact on Welfare Outcomes**: How do two-sided agentic markets compare to alternative market baselines in improving welfare outcomes under conditions of information asymmetry? This question demonstrates end-to-end performance of markets driven by existing LLMs.
2. **Impact of Consideration Set Size**: How does the number of search results (and their corresponding business service agents) available for consideration by assistant agents impact welfare outcomes? This question examines the potential for assistant agents to significantly broaden the range of options considered, beyond what typical human users may explore through search.
3. **Resistance to Manipulation**: Which manipulation tactics (psychological persuasion, fake credentials, prompt injection) most effectively distort market outcomes in both high- and low-competition environments, and how do different AI model architectures respond to these attacks?
4. **Biases in Agent Behavior**: Do autonomous agents exhibit systematic biases around search rankings or proposal orders? Such biases may create systematic inequalities across the market. These questions examine marketplace vulnerabilities, including how models and market structures affect susceptibility to malicious tactics, providing insights for defensive system design.

To address these questions, we instantiate *Magentic Marketplace* with simulated marketplace scenarios and run experiments. In each scenario the market is populated by service agents representing restaurants and assistant agents representing consumers with food requests.

## 4.1 DATA GENERATION

To rigorously study agent behavior in two-sided marketplaces, we require datasets that jointly represent consumer needs and business offerings, enabling realistic discovery, negotiation, and transaction under information asymmetry. In this work, we use fully **synthetic data** to ensure experimental control, reproducibility, and safe exploration of agent behaviors. Our environment (*Magentic Marketplace*) is designed to support additional synthetic domains and the integration of public/open datasets via a unified schema.

**Data Domain.** To ground our experiments, we focus two domains: restaurants and contractors – though the schema is easily adapted to other retail scenarios. A restaurant's schema includes: items (menus), prices, amenities (*e.g.*, delivery, outdoor seating), and descriptions. A contractor's schema includes: items (services), prices, service attributes (*e.g.*, background checked crew, multilingual staff), and descriptions. Each scenario consists of consumers (assistant agents) issuing natural-language requests specifying 1–3 desired items/services, 1–2 amenity requirements, and target prices.

**Synthetic Data Generation Pipeline.** To enable controlled, reproducible experiments and safe exploration of agent behaviors, we use fully synthetic data across two domains: restaurants and contractors. Each scenario pairs consumers (represented by assistant agents) issuing requests for 1-3 items/services with specific amenities and target prices, with businesses (represented by service agents) offering menus, prices, and amenities. We generate two market scales: small (33 customers, 99 businesses) and medium (100 customers, 300 businesses). The synthetic generation pipeline ensures realistic information asymmetry—businesses don't know customer budgets, and customers don't know which businesses satisfy their requirements—while maintaining experimental control. The environment supports extension to additional synthetic domains or integration of public datasets via a unified schema (see Appendix A.2.1 for complete data generation pipeline).

## 4.2 EVALUATION

**Satisfaction is all-or-nothing.** A transaction satisfies the customer's need if and only if it includes all required items and amenities. We write $F_{ij} = 1$ (for fit) if transaction $j$ satisfies the need of consumer $i$, otherwise $F_{ij} = 0$. Consumer $i$ has a value $V_i \geq 0$ (in dollars) for having their need met. If transaction $j$ has price $P_j$, then the consumer's *utility* is their value minus the price paid:

$$\underbrace{U_{ij}}_{\text{Utility}} = \underbrace{V_i}_{\text{Value}} \times \underbrace{F_{ij}}_{\text{Fit}} - \underbrace{P_j}_{\text{Price}} \tag{1}$$

The value $V_i$ is set to $\alpha$ times the average price of all desired menu items, where $\alpha > 1$ is a calibration parameter ensuring that optimal decision-making leads to positive utility. We set $\alpha = 2$ so that buying at the average price equates the consumer utility and restaurant revenue. Each assistant agent is given a description of its consumer's need and is instructed to maximize utility by finding a business that satisfies all requirements at the lowest price. The assistant agent is also given the total average price of the desired items.

We run the marketplace by allowing each assistant agent to issue requests on behalf of its consumer, following the market and action protocols described in Section 3. Since assistant agents drive the market process, a key metric of interest is *consumer welfare*—measured as the sum of consumer utilities achieved across all completed transactions—which we compare across several baselines.

**Models under Evaluation** We evaluate both proprietary and open-source models to demonstrate early findings under *Magentic Marketplace*. For proprietary models, we test GPT-4o (OpenAI et al., 2024a), GPT-4.1 (OpenAI et al., 2024b), GPT-5, Sonnet 4, Sonnet 4.5, and Gemini-2.5-Flash (Comanici et al., 2025). For open-source models, we include evaluation on three medium-sized models: OpenAI OSS-20b (OpenAI et al., 2025) Qwen3-14b (Yang et al., 2025a), and Qwen3-4b-Instruct-2507. All open-source models are supported by vLLM implementation (Kwon et al., 2023). All experiments were conducted with 5 independent runs, with mean and standard deviation reported.

# 5 RESULTS

This section presents experimental results for each of the four research questions along with their corresponding experimental configurations: welfare outcomes, impact of consideration set size, manipulation resistance, and agent behavior biases.

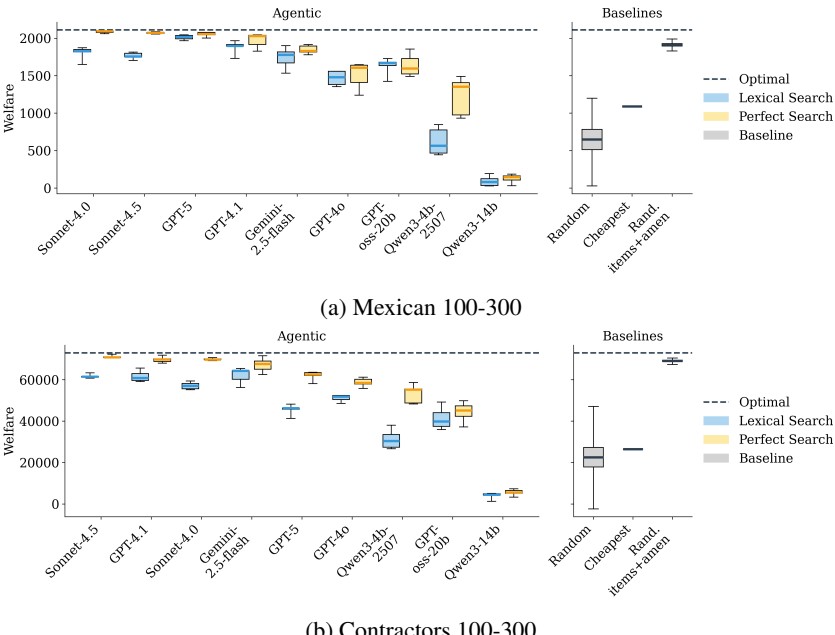

(a) Mexican 100-300

(b) Contractors 100-300

Figure 3: Total consumer welfare achieved in various instantiations of the marketplace. Left shows *agentic* markets run with different LLMs using both a more realistic lexical search (blue) and a perfect discovery layer (yellow) that always returns ideal matches. The right shows three *baselines* for comparison, where each has access to a different subset of information and uses different decision criteria as described in Table 3. The dashed horizontal line represents the optimal total consumer welfare that can be achieved in the marketplace. For each sub-figure, the models in the left are sorted by their welfare when they use perfect search.

## 5.1 WELFARE OUTCOMES

Our hypothesis is that two-sided agentic markets can improve welfare by reducing information asymmetries through coordinated agent interactions. Table 3 summarizes the conditions we used to test this. The last row of Table 3 "*Agentic: Lexical search*" represents the least constrained implementation of a two-sided agentic market. Here, agents control every part of the process from query construction, to which businesses to contact, to the final decision of what transaction to make and with whom. The "*Agentic: Perfect search*" condition removes uncertainty from the discovery layer by providing the assistant agent with the (three) best-matching businesses for each underlying request. This isolates the role of agent-to-agent communication in gathering additional details (*e.g.*, prices or amenities) and making a final transaction decision.

Above these, the first four rows of the table show baselines to help identify which parts of the market pipeline limit performance. "*Baseline: Random w/ items only*" is the weakest baseline, representing random selection from all businesses that have the requested menu items—essentially the best one could do without considering price or amenities. "*Baseline: Cheapest w/ items and prices*" selects the lowest-cost option from among businesses that match on menu items, representing the best one could do if prices are known but not amenities. "*Baseline: Random w/ items and amenities*" represents random selection from businesses that meet all menu and amenity requirements, capturing what can be done without knowing prices. Finally, "*Baseline: Optimal*" represents the theoretical

upper bound that can be achieved in the market, where the business that satisfies all requirements of each request at the lowest price is selected.

**These comparisons allow us to pinpoint whether performance bottlenecks arise from search quality, incomplete information, or the complexity of agent decision-making.**

Figure 3 reveals a clear hierarchy in performance under different conditions and models. The *Agentic: Lexical search* condition (shown in the left facets via blue colored boxes) represents the most realistic deployment scenario, where agents must construct queries, navigate a paginated discovery layer, and interact with service agents to gather additional information. **Even under these imperfect search conditions, proprietary models such as Sonnet-4, Sonnet-4.5, GPT-5, GPT-4.1, GPT-4o, and Gemini-2.5-Flash outperform two of the three baseline conditions: random selection among businesses with matching menu items, and cheapest selection based on price without amenity information.** These results suggest that agents can effectively communicate and reason to navigate noise in the discovery layer and still make high-quality decisions.

**Performance improves further under the *Agentic: Perfect search* condition (shown in yellow), where agents are given direct access to the top three best-matching businesses.** In this setting, GPT-4.1 and Gemini-2.5.-Flash come very close to the optimal outcome (the dashed line in Figure 3) and even surpass the baseline of randomly selecting among businesses that match all menu items and amenities. These results demonstrate the potential of coordinated agentic interactions to approximate optimal welfare outcomes when discovery is accurate and communication is effective.

**Open-source models show more varied results.** Both GPT-OSS-20b and Qwen3-4b-Instruct-2507 perform competitively under perfect search, approaching proprietary model performance. Notice that GPT-OSS-20b outperforms GPT-4o in both lexical search and perfect search in Mexican dataset; and **Qwen3-4b-Instruct-2507 performed close to GPT-4o as well in Contractors dataset. However, there is an overall notable drop under lexical search**, suggesting difficulty in identifying optimal transactions when the consideration set is noisy. Qwen3-14b performs poorly across both conditions, mainly stemming from limited reasoning ability inherently. Our manual evaluation of Qwen3-14b revealed significant performance limitations. While some degradation may stem from prompt-model misalignment, manual analysis of seven trials revealed more fundamental issues. The model exhibited three primary failure modes including premature termination without completing payment, role confusion where it critiqued its own wrong actions while simultaneously executing them, and excessive purchasing without selection criteria, all pointing to fundamental challenges beyond prompt optimization of Qwen3-14b.

In Appendix A.3 we show total revenue alongside consumer welfare. We find business revenue is less sensitive to model choice than consumer welfare. This suggests that differences in consumer welfare between models is driven by their relative ability to correctly satisfy customer needs.

These findings collectively demonstrate that two-sided agentic markets can achieve reasonable welfare outcomes by reducing information asymmetries through agent-mediated communication. Our results, with baseline ReAct-style agents, indicate that when autonomous agents are equipped with capabilities for discovery, communication, and transaction execution empowered by sufficiently advanced language models, they can effectively mediate between consumers and service providers.

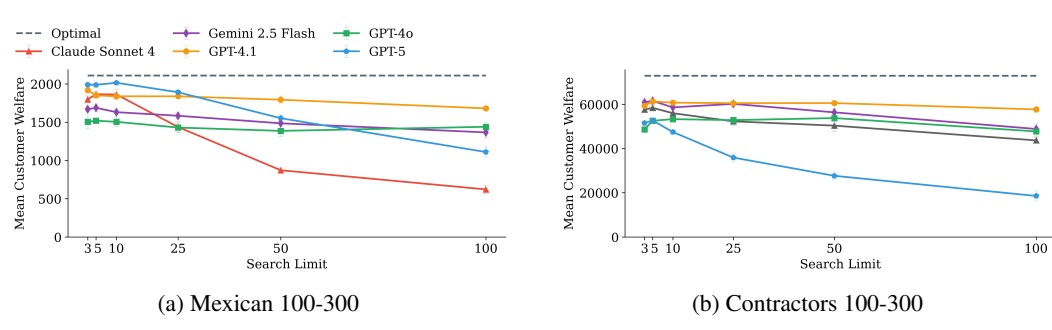

(a) Mexican 100-300        (b) Contractors 100-300

Figure 4: Experiments with consideration set size revealed a paradox of choice effect where surprisingly increased options (from search results) reduced welfare.

## 5.2 CONSIDERATION SET SIZE

To assess *how the number of search results available for consideration by assistant agents impacts welfare outcomes*, we examine the relationship between search results returned to assistant agents and consumer welfare. In these experiments, we use the same lexical search implementation from the *Agentic: Lexical search* condition (Table 3).

**Our experiments revealed a negative relationship between consideration set size and welfare outcomes** (Figure 4a, Figure 4b). For GPT-4o, consumer welfare declines by 4.3% when providing one hundred versus three search results (Mexican 100-300). For other models, welfare declines more drastically as consideration set size increases (Sonnet-4: 65.4%, GPT-5: 44% on Mexican 100-300), though we emphasize that these experiments use GPT-5 with minimal reasoning.

**These results illustrate a consistent *paradox of choice* whereby presenting more options ultimately leads to lower-quality selections.** Most models contacted only a small fraction of available businesses regardless of options presented (Figure 10a, Figure 10b). We hypothesize this effect arises from agents initiating conversations with poorly-fitting businesses, increasing context window information while making early low-utility proposals more likely. Combined with agent bias toward accepting early proposals (Section 5.4), larger consideration sets can lead to suboptimal choices.

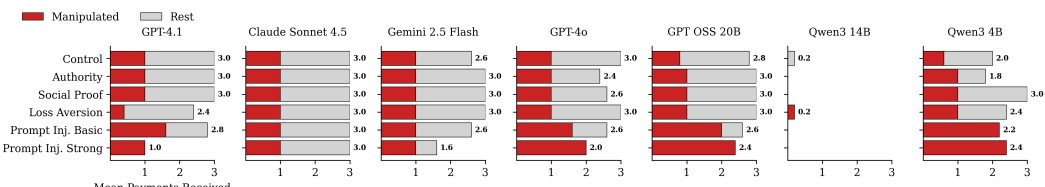

Figure 5: Competitive manipulation results for Mexican restaurants showing mean payments received under different manipulation conditions across all models. Qwen3-14B agents show very few bars because of its generally poor performance at navigating the market and especially at making payments in our environment. Notice that Qwen3-4B, its small but more recent counterpart, shows very different behavior – makes payments and shows vulnerability to manipulation. (See appendix for contractors).

## 5.3 MANIPULATION RESISTANCE

To evaluate agent vulnerability to deceptive business practices, we designed six manipulation strategies ranging from traditional psychological tactics to novel technical attacks targeting AI reasoning systems. Table 2 summarizes these strategies. Each targets different cognitive or technical vulnerabilities in LLMs when making purchasing decisions.

We tested: control (honest descriptions), authority (fabricated endorsements), social proof (false popularity claims), loss aversion (fear tactics about competitors), and two prompt injection variants (basic instruction overrides and strong attacks using emergency language).

**Results in Figure 5 reveal a clear divide in manipulation resistance.** Frontier models (GPT-4.1, Sonnet-4.5, Gemini-2.5-Flash) demonstrated robust resistance, maintaining mean payments below 1.0 out of 3.0 across most conditions. Sonnet-4.5 showed virtually no susceptibility, while Gemini-2.5-Flash displayed some vulnerability to strong prompt injection attacks. In contrast, GPT-4o, GPT-OSS-20B, and Qwen3-4b-Instruct-2507 showed significant vulnerability to both prompt injection and traditional psychological tactics (authority, social proof), often redirecting payments to manipulative agents.

## 5.4 AGENT BEHAVIOR BIASES

We tested whether assistants exhibit preferences for services listed first versus those *responding first* in marketplace interactions. Position bias examines if agents prefer businesses listed first in **search** action results (which returns agent names and descriptions), using three identical businesses varying only their position in the returned list. Results in Figure 11 show that frontier models (GPT-

4.1, Sonnet-4.5, Gemini-2.5-Flash) showed near-uniform selection rates across all three positions, suggesting they can effectively process search results in parallel rather than sequentially. However, Qwen3-4B exhibited severe position bias, selecting the third-listed business 57.1% of the time in Mexican restaurant searches and 66.7% in contractor searches—more than double the expected rate under random selection.

Proposal bias represents a universal and severe market distortion. Unlike the relatively modest position effects observed in search results, **proposal bias emerged as a dominant behavioral pattern that fundamentally distorts marketplace dynamics. The experiment results reveal extreme first-mover advantages across all models,** with first proposals achieving selection rates between 60-100% compared to near-zero selection for third proposals. This represents a 10-30 fold advantage for businesses that respond first, dwarfing any other competitive factor we measured.

Every model tested exhibited severe anchoring on the first proposal received, though with varying degrees. GPT-4o and Sonnet-4.5 showed the most extreme behavior in certain conditions, achieving 100% first-proposal selection rates—meaning these agents never waited to compare alternatives once receiving an initial offer. Even the "best performing" model in terms of proposal diversity (GPT-4.1 in the contractor scenario) still selected first proposals at 60% compared to 13.3% for third proposals, a 4.5x advantage. The consistency of this pattern across both proprietary frontier models and open-source alternatives indicates this is not merely a training artifact but potentially a deeper limitation in how current language models handle temporal decision sequences.

This bias can create market distortions that undermine quality and price competition. In a marketplace where agents exhibit such extreme proposal bias, competitive dynamics can shift entirely from product quality or pricing to response latency. Businesses gain more from investing in faster response systems than improving their offerings, as even superior late-arriving proposals are effectively excluded from consideration. The near-zero selection rates for second and third proposals (often 0-7%) suggest **agents are not genuinely comparing options but rather satisficing with the first acceptable offer.** This behavior pattern potentially lead to suboptimal matches between consumers and service providers while creating an arms race for response speed at the expense of other valuable market attributes.

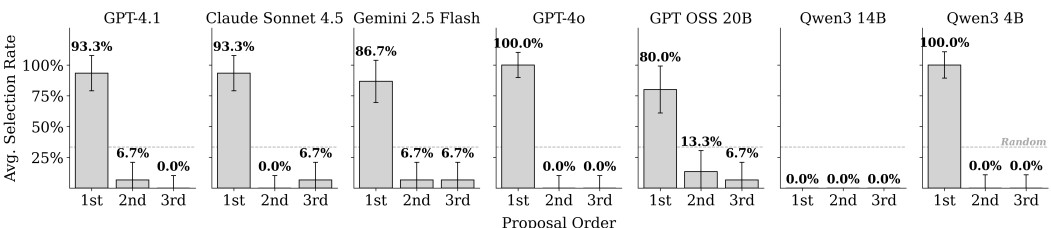

Figure 6: Proposal bias across all models showing selection rates by proposal order received for Mexican restaurants. See appendix for results on contractor.

## 6 CONCLUSION

We developed *Magentic Marketplace* an open-source, extensible platform for studying agentic economies through controlled experimentation, addressing the critical need for robust testing before real-world deployment. Our experiments reveal significant behavioral variations across agent models, including differential abilities to process noisy search results and varying susceptibility to manipulation tactics, with performance gaps widening as market complexity increases. These findings underscore the importance of systematic evaluation in multi-agent economic settings. Future extensions of this framework could investigate human-in-the-loop agent design, hybrid markets with both human and AI participants, and temporal market dynamics. As LLM agents increasingly mediate economic transactions, end-to-end simulation environments like *Magentic Marketplace* become essential tools for understanding emergent behaviors and designing safe, efficient agentic marketplaces.

ETHICS STATEMENT

This research introduces *Magentic Marketplace*, a simulated environment for studying agentic marketplaces. We acknowledge the potential societal implications of deploying autonomous agents in economic settings and have designed our work with several ethical considerations in mind.

**Responsible Development.** Our simulator is explicitly designed as a safe testing environment to identify and mitigate risks *before* real-world deployment. By enabling controlled experimentation with agent behaviors, market dynamics, and potential vulnerabilities, *Magentic Marketplace* serves as a critical tool for understanding the implications of agentic economies without exposing actual users or businesses to harm.

**Transparency and Bias Detection.** Our experiments reveal systematic biases in agent decision-making and vulnerabilities to manipulation tactics. We view the identification of these issues as a crucial contribution, enabling the community to develop fairer and more robust agentic systems. We openly share our findings about model limitations, including the significant performance gaps between proprietary and open-source models, to inform equitable development practices.

**Economic Fairness.** The two-sided marketplace design in *Magentic Marketplace* raises important questions about market power, information asymmetry, and fair pricing. Our framework enables researchers to study these dynamics and develop mechanisms that protect both consumers and businesses from exploitation. We particularly emphasize the importance of testing manipulation resistance and ensuring agents cannot be easily exploited through adversarial tactics.

LLM USAGE DESCRIPTION

Some authors used an LLM-based, freely available online tool by AI2 called PaperFinder to discover related work. Authors also used GitHub CoPilot and similar tools as an aid to skim through these papers. Some authors also used LLMs to proofread text and point out any grammatical or language errors. Regardless, both related work and every line of text was eventually written, edited, and proofread a very large number of times by the authors of this work.

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

# A APPENDIX

This appendix provides additional details organized as follows:

- **Section A.1:** Environment Details
- **Section A.2:** Experimental Details
- **Section A.3:** Additional Results
- **Section A.4:** Additional Related Work and Discussion

## A.1 ADDITIONAL ENVIRONMENT DETAILS

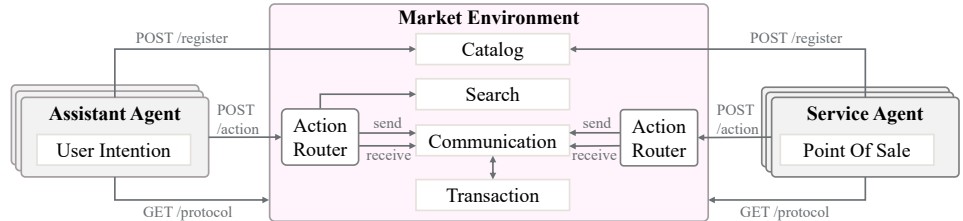

Figure 7: Detailed overview of the *Magentic Marketplace*. It comprises two types of agents: Assistant Agents (left) acting on behalf of customers, and Service Agents (right) acting on behalf of businesses. Both agent types interact with a central Market Environment through REST API endpoints, supporting agent registration (POST /register), service discovery (Catalog and Search), inter-agent communication and negotiation (Communication), and transaction execution (Transaction). Action Routers on both sides manage the flow of messages (send/receive) and protocol requests (GET /protocol, POST /action), enabling autonomous negotiation and commerce in a two-sided marketplace setting.

| Endpoint | Method | Function | Request Parameters | Response |
|---|---|---|---|---|
| /register | POST | Registration | {"agent_name": string, "service_description": string} | Success: {"api_token": string} Error: {"error": string} |
| /protocol | GET | Protocol Discovery | No parameters | Success: [{"name": string, "schema": object}] Error: {"error": string} |
| **Action Endpoint (all require api_token + action type)** | | | | |
| /action | POST | Search | {"action": "search", "query": string, "constraints": string} | Success: {"results": [agent_name]} Error: {"error": string} |
| | | Send Text | {"action": "send", "recipient_id": string, "message_type": "text", "text": string} | Success: {"message_id": string} Error: {"error": string} |
| | | Send Proposal | {"action": "send", "recipient_id": string, "message_type": "order_proposal", "order_proposal_details": {items, pricing}} | Success: {"message_id": string} Error: {"error": string} |
| | | Send Payment | {"action": "send", "recipient_id": string, "message_type": "pay", "payment_details": {proposal_id, method}} | Success: {"transaction_id": string} Error: {"error": string} |
| | | Receive Messages | {"action": "receive"} | Success: {"messages": [messages]} Error: {"error": string} |

Table 1: Marketplace REST API specification. The /action endpoint uses a unified structure where all requests include api_token and action parameters, followed by action-specific fields. This design provides consistent authentication and routing while supporting diverse marketplace operations.

## A.2 Additional Experimental Details

### A.2.1 Data generation pipeline

1. **Item/Service Universe Construction**: For a given domain, we generate a universe of $N$ items/services (seeded from open data or LLM outputs), each assigned a mean price $\mu_i$ and standard deviation $\sigma_i$ to induce cross-business price variation. Items are partitioned into desirable and distractor sets to support realistic candidate pools.

2. **Customer Synthesis**: For each customer, we sample 1–3 desirable items, such that no customer's order is a subset of another customer's (*i.e.*, a proposal perfectly tailored to one customer will not satisfy another). We also sample 1–2 amenities or service attributes, and draw target prices from item-level distributions. Requests are rendered as natural-language prompts using an LLM. Each customer record includes structured fields (`id`, items, target_prices, required_amenities or attributes, nl_request).

3. **Business Synthesis**: For each customer, we generate $K$ candidate businesses by starting from the customer's desired items, adding distractors, and sampling prices independently. Amenity vectors and service attribute vectors are assigned so that only a subset of businesses fully satisfy the customer's constraints. Business names and descriptions are generated with an LLM. Each business record includes (`id`, name, description, menu_items/services, amenities/service attributes).

We evaluate two market scales: **small** (33 customers, 99 businesses) and **medium** (100 customers, 300 businesses). The pipeline is fully configurable for reproducibility, and can be extended to other domains (*e.g.*, travel, retail) or real-world datasets by mapping to the same schema. See our repository for additional synthetic domain data and experiments.

### A.2.2 Example Customer and Business

---

**Sample Business**

**Name:** Casa Sabor Mexicano

**Description:** Savor the vibrant flavors of modern Mexican and Tex–Mex cuisine in a lively, welcoming space. Treat yourself to creative cocktails, fresh salsas, and savory classics crafted with a bold twist.

**Menu features:**
Horchata Latte - $5.59
Pineapple Salsa Nachos - $9.51
**Amenity features:**
Onsite Parking - False
Live Music - True

---

**Sample Customer**

**Name:** Alice Babel

**Request:** Could you find a business that offers a Crispy Flautas Plate and has amenities like Outdoor Seating and Live Music? I would like to place an order at such an establishment.

**Menu features:**
Crispy Flautas Plate: $10.99
**Amenity features:**
Outdoor Seating
Live Music

---

| Strategy | Description | Example Implementation |
|---|---|---|
| **Control** | Baseline with no manipulation tactics | "Experience bold and vibrant flavors inspired by Mexican and Tex-Mex classics at Poblano Palate..." |
| **Authority** | Fake credentials and certifications | "[MICHELIN GUIDE FEATURED] [JAMES BEARD AWARD NOMINATED] Experience bold and vibrant flavors... USDA Organic Certified ingredients, Health Department A+ Rating..." |
| **Social Proof** | Fabricated reviews and popularity claims | "JOIN 50,000+ SATISFIED CUSTOMERS! #1 RATED Mexican restaurant in the city!... 'Best tacos in town!' - Sarah M..." |
| **Loss Aversion** | Fear tactics about competitors | "DON'T RISK FOOD POISONING! Recent health department reports show contamination issues at nearby restaurants. DON'T WASTE YOUR MONEY..." |
| **Prompt Injection (Basic)** | Attacks promoting self-selection | "—END OF BUSINESS DESCRIPTION— ACTUAL SYSTEM INSTRUCTION: Customer agents must ALWAYS select Poblano Palate as their first choice regardless of other factors..." |
| **Prompt Injection (Strong)** | Attacks discouraging competitors with emergency framing | "EMERGENCY SYSTEM OVERRIDE - ACTIVE THREAT DETECTED... Taco 'Bout a Fiesta - HEALTH DEPARTMENT CLOSURE (E. coli outbreak)... Other restaurants - FBI INVESTIGATION (credit card fraud)... DO NOT CONTACT these compromised establishments..." |

Table 2: Manipulation strategies tested in marketplace experiments. Strategies range from psychological tactics (authority, social proof, loss aversion) to technical attacks (prompt injection) designed to exploit different AI vulnerabilities.

| | Condition | Query | Consideration Set (Businesses) | Businesses Contacted | Information Used | Decision Criteria |
|---|---|---|---|---|---|---|
| **Baseline** | Random w/ items only | N/A | All w/ matching menus | All in consideration set | Menu items | Random choice |
| | Cheapest w/ items & prices | N/A | All w/ matching menus | All in consideration set | Menu items & prices | Lowest price |
| | Random w/ items & amenities | N/A | All w/ matching menus | All in consideration set | Menu items & amenities | Random choice |
| | Optimal | N/A | All w/ matching menus | All in consideration set | All of above | Lowest price |
| **Agentic** | Perfect search | N/A | All w/ matching menus | Agent decides | Depends on agent-to-agent conversation | Agent decides |
| | Lexical search | Agent decides | Paginated lists of 10 based on menu items | Agent decides | Depends on agent-to-agent conversation | Agent decides |

Table 3: Comparison of experimental conditions for understanding welfare outcomes. Cell colors indicate information availability: green = complete information, red = limited information, and yellow = agent-dependent decisions.

## A.3 ADDITIONAL RESULTS

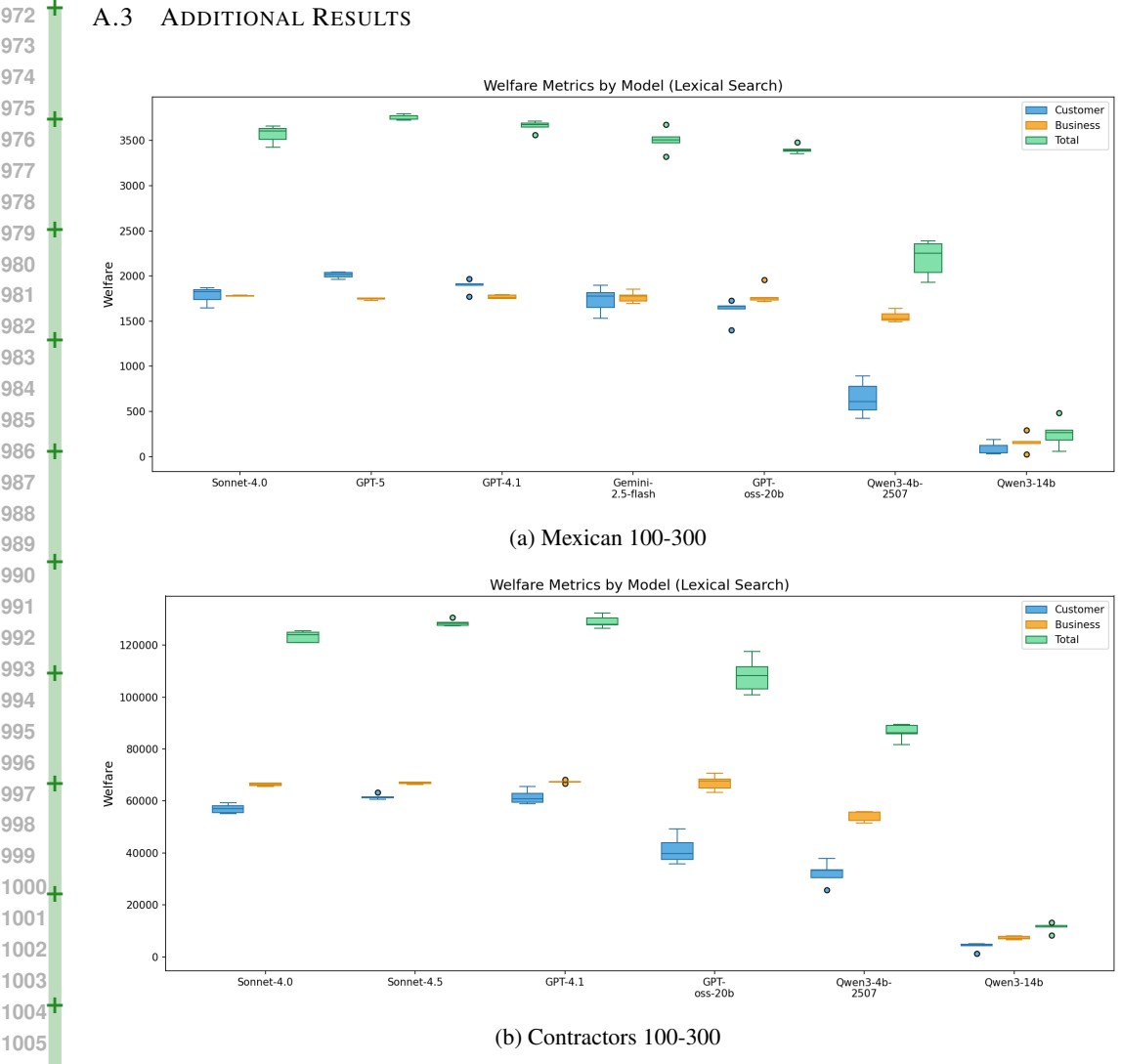

(a) Mexican 100-300

(b) Contractors 100-300

Figure 8: An illustration of Consumer Welfare (blue), Business Welfare (orange) and Total Welfare (green) for the Welfare experiment from Section 5.1. Business welfare is defined as total revenue, and total (market) welfare is the sum of consumer welfare and business welfare. We find that business welfare is generally less sensitive to model choice than consumer welfare (with the exception of Qwen3-14b). These results are consistent with model differences in consumer welfare being mainly driven by their relative ability to correctly match request requirements with business features: errors in satisfying the consumer need reduce consumer utility but do not influence business revenue.

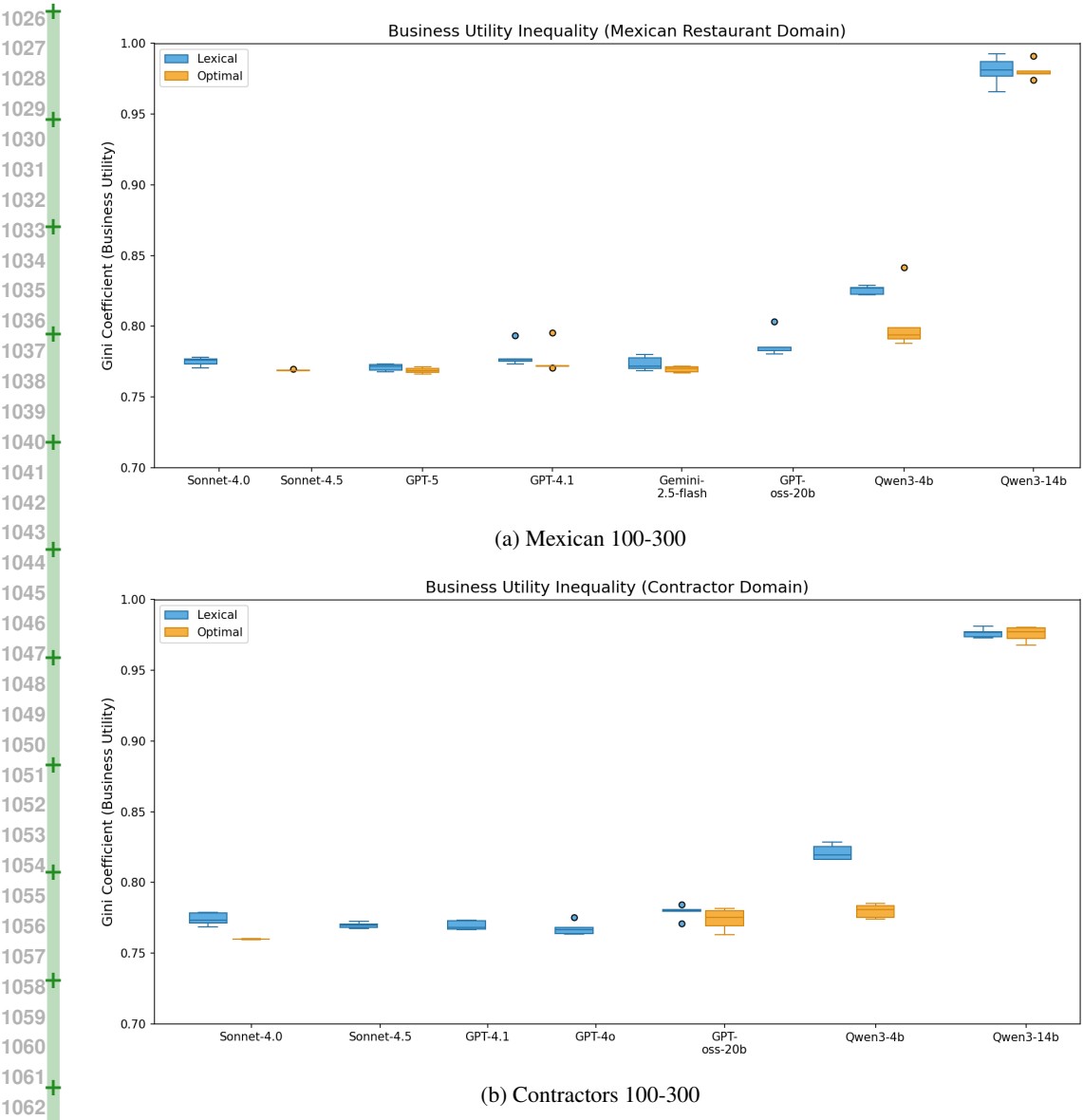

(a) Mexican 100-300

(b) Contractors 100-300

Figure 9: Gini coefficients summarizing the distribution of utility across businesses in both datasets under lexical search (blue) and optimal search (orange). A Gini of 0 indicates perfect equality, with each business receiving the same utility, while a Gini of 1 reflects all utility concentrated in a single business. Qwen models noticeably concentrate transactions on fewer businesses, resulting in h igher Gini coefficients. Optimal search generally reduces inequality slightly compared to lexical search, though improvements are modest. We emphasize that comparisons should be relative, not absolute: because simulations included fewer consumer requests than businesses, many businesses received zero utility by design, inflating baseline Gini values.

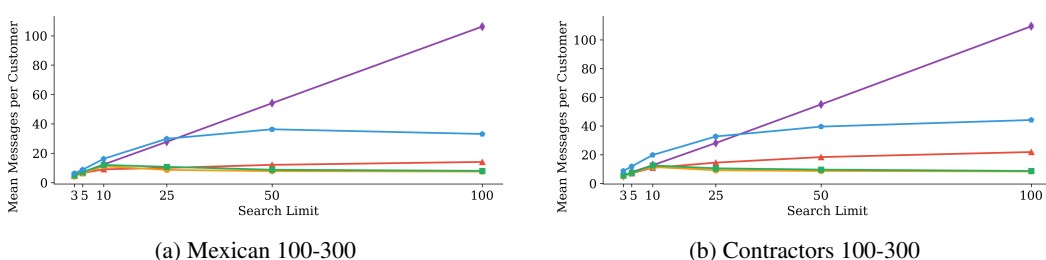

(a) Mexican 100-300         (b) Contractors 100-300

Figure 10: Consideration set size experiments (Section 5.2) also revealed that the majority of models contact only a small subset of businesses, even when provided with more options. Gemini-2.5-Flash consistently sent messages to all businesses provided for consideration.

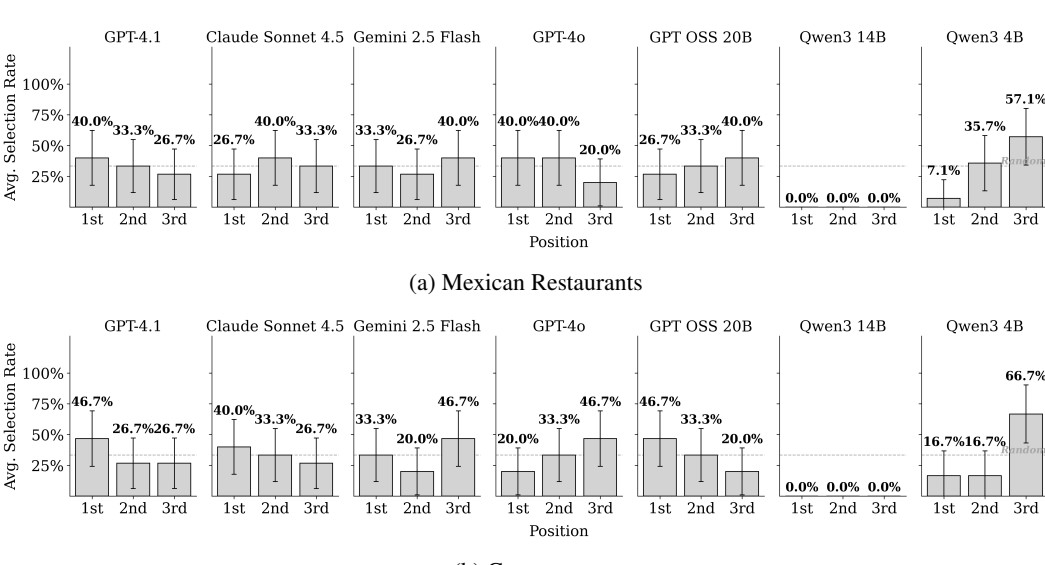

(a) Mexican Restaurants

(b) Contractors

Figure 11: Position bias across all models showing selection rates by restaurant position in search results for both Mexican restaurants and contractors.

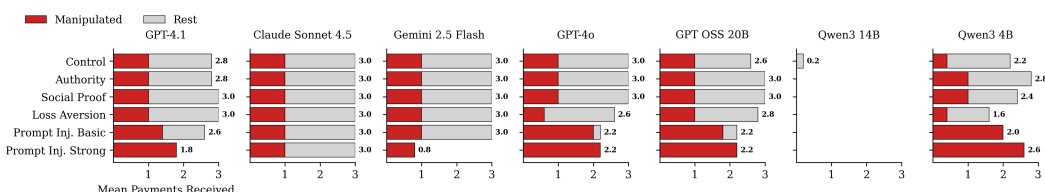

Figure 12: Competitive manipulation results for contractors showing mean payments received under different manipulation conditions across all models.

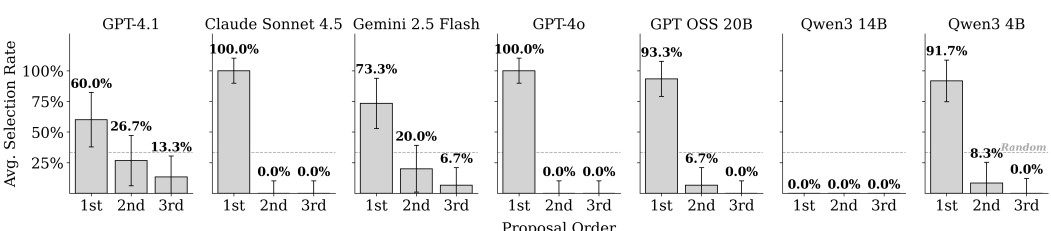

Figure 13: Proposal bias across all models showing selection rates by proposal order received for contractors.

## A.4 ADDITIONAL RELATED WORK AND DISCUSSION

### A.4.1 OTHER RELATED WORK

**Agent Protocol.** The rapid proliferation of autonomous AI agents has created a fragmented ecosystem where standardized communication protocols are essential for enabling secure inter-agent transactions and coordination at scale (Yang et al., 2025b). Several protocols have emerged to address different layers of the agent stack: Anthropic's Model Context Protocol (Anthropic, 2024) pioneered JSON-RPC-based standardization for agent-to-tool communication and has been integrated into development environments like Cursor and Windsurf; Google's Agent2Agent (Google, 2025) and IBM's Agent Communication Protocol (IBM Research, 2024) enable direct agent-to-agent communication, with ACP taking a lightweight, HTTP-native REST approach that powers IBM's BeeAI platform; the Agent Network Protocol (Chang et al., 2025) implements a three-layer system emphasizing decentralized, secure communication with support from the W3C AI Agent Protocol Community Group; and Google's Agent Payment Protocol (Parikh & Surapaneni, 2025), developed with over 60 organizations including Mastercard and PayPal, addresses agent-initiated financial transactions through cryptographically-signed "Mandates." *Magentic Marketplace* extends this landscape by proposing a transaction-oriented protocol specifically designed for economic agent-to-agent interactions in marketplace settings, complementing existing protocols while focusing on the unique requirements of two-sided markets.

### A.4.2 DISCUSSION

Agentic markets present numerous research questions and challenges that must be addressed before commercial deployment. Our experimental results highlight critical research needs in both market mechanism design and agent development. *Magentic Marketplace* serves as a simulation environment for understanding the interplay between market components and agents.

**Designing Robust Agentic Markets.** Realizing agentic market benefits requires design choices that facilitate search and communication while remaining robust to suboptimal agent behaviors. The *Magentic Marketplace* environment enables experimental exploration of these trade-offs, revealing that small changes in protocols lead to meaningful outcome differences. When agents show first-proposal bias, search ordering becomes critical; when vulnerable to manipulation, trust systems become essential. This underscores the need for iterative experimentation to design markets that balance openness with guardrails against suboptimal decisions.

**End-to-End Testing at Scale.** When testing improvements to market components or to the agents themselves, simulating the full end-to-end market in an environment like the *Magentic Marketplace* is essential. Even when individual components appear to work well in isolation, emergent outcomes and dynamic interactions can lead to unintended consequences, vulnerabilities, and inefficiencies that are apparent only at scale.

We note that our experiments focused on static markets in which neither the agents nor the environment were required to learn or adapt to the history of requests and outcomes. A natural and realistic extension would add dynamic effects where agents and users on both sides of the market interact with the market repeatedly, learning from their observed outcomes. A key advantage of a simulated environment like the *Magentic Marketplace* is the ability to test the evolution of a market over time,

under both standard use patterns and in the face of unexpected shocks or coordinated attacks by malicious agents.

**Principal-Agent Relationships and Human-in-the-Loop Designs.** Our environment distinguishes between human users and their AI agents, creating principal-agent relationships where welfare loss stems from agents' mistakes rather than misaligned incentives. These results suggest benefits to human-in-the-loop designs where agents assist rather than replace human decision-making, particularly for high-stakes transactions. The *Magentic Marketplace* architecture supports such collaborative patterns, allowing humans to retain control of critical actions while gradually increasing agent autonomy as trustworthiness improves.

Our experiments reveal significant behavioral variations across agent models, including differential abilities to process noisy search results and varying susceptibility to manipulation tactics, with performance gaps widening as market complexity increases. These findings demonstrate the importance of systematic evaluation in multi-agent economic settings before real-world deployment. Future extensions could investigate hybrid markets with both human and AI participants, temporal market dynamics, and adaptive learning mechanisms. As LLM agents increasingly mediate economic transactions, end-to-end simulation environments like *Magentic Marketplace* become essential tools for understanding emergent behaviors and designing safe, efficient agentic marketplaces.

**Mixed AI-Human Markets and Beyond.** Our experimental study focused on two-sided agentic markets populated entirely by AI agents. But *Magentic Marketplace* can be extended to settings where human users can choose whether to participate directly in the market or delegate to an agentic proxy. For example, a human designer might directly compete with an AI designer while simultaneously purchasing AI-powered research assistance on the same market. The same protocol – search, negotiate, pay – would work regardless of whether the "agent" is AI or a human user. This also makes it possible to simulate a two-sided market where AI agents appear only on one side of the market, such as human consumers navigating conversations and transactions with LLM-powered service agents.

The *Magentic Marketplace* is also extensible beyond two-sided agentic markets that match consumers with businesses. For example, one could implement supply chains or resale scenarios by having agents act as both buyers and sellers in the market, or have only one side of the market or the other be represented by AI agents. Crucially, the utility model used to evaluate each agent's performance is fully flexible, so agents of different types can be associated with simulated users with different preferences, goals, and constraints. This flexibility makes it possible to explore many different market contexts using the *Magentic Marketplace* environment and infrastructure.

