# OpenReview forum: "Magentic Marketplace: An Open-Source Environment for Studying Agentic Markets"
_ICLR.cc/2026/Conference — Submitted to ICLR 2026_

### Official Review · Reviewer_JBMT · 2025-10-23

**Soundness:** 2
**Presentation:** 2
**Contribution:** 2
**Rating:** 2
**Confidence:** 3

**Summary:**

The paper presents *Magentic Marketplace*, a simulation framework for studying two-sided markets in which AI agents interact on behalf of consumers and providers. Using this framework, the authors analyze how different market conditions and underlying AI models affect consumer welfare, behavioral biases, and susceptibility to manipulation.

**Strengths:**

The paper explores a timely and significant topic at the intersection of economic behavior and AI ecosystems, offering practical tools to study both the potential and the limitations of AI-driven economies. It effectively motivates its perspective and provides a well-grounded background for the reader. From an environment design standpoint, I find the paper’s approach particularly valuable: extending beyond the stylized setups commonly found in the literature (such as those focusing solely on negotiation stages) and instead introducing a comprehensive pipeline that integrates multiple types of interactions. Such an approach is essential for understanding how real-world platforms may operate when AI agents act on both sides of the market.

**Weaknesses:**

1. The paper focuses exclusively on consumer welfare, yet the setting is inherently a two-sided market. I would expect to see a corresponding analysis of the welfare of business providers as well. Considering both sides is essential for capturing the full set of economic trade-offs and for understanding how market design choices affect overall economic efficiency.

2. The paper would benefit from providing more practical insights into how specific design choices shape economic outcomes. For example, are there design parameters that could steer the outcome toward outcomes more favorable to consumers or providers? While the discussion around search mechanisms (lexical vs. perfect search) offers a valuable start, there appear to be many additional degrees of freedom worth exploring, such as how agent communication is structured (natural language vs. formal protocols) or how supply and demand dynamics are modeled. The absence of such analysis makes the current exploration feel somewhat limited in scope.

3. The term “information asymmetry” appears several times but is not clearly defined in context. It remains unclear what constitutes the private information, who possesses it, and how its presence affects the resulting economic outcomes. A more precise definition and explicit modeling of these asymmetries would strengthen the conceptual clarity of the paper.

4. Although the authors themselves note in the ethical statement that fairness is a key concern in the context of two-sided market design, the paper does not provide a concrete definition or formal treatment of fairness. I would expect to see a clear definition of what fairness means in this setting, and at least some preliminary analysis of how the proposed mechanisms perform with respect to this criterion.

**Another minor comment:** Some citations seem to specify the incorrect order of authors, for instance:

Moshe Tennenholtz, Eilam Shapira, Omer Madmon, and Roi Reichart -> Eilam Shapira, Omer Madmon, Roi Reichart, and Moshe Tennenholtz (Can llms replace economic choice prediction labs? the case of language-based persuasion games)

**Questions:**

I do not have any specific questions, but I am willing to hear your thoughts and comments on my concerns.

---

> ### Author Response · Authors · 2025-11-24
>
> >  The paper focuses exclusively on consumer welfare, yet the setting is inherently a two-sided market. I would expect to see a corresponding analysis of the welfare of business providers as well. Considering both sides is essential for capturing the full set of economic trade-offs and for understanding how market design choices affect overall economic efficiency.
>
> **For our first welfare experiments**, we focus on how the model choice, available information, and consideration set available to the assistant agent influence consumer outcomes. Since **these experiments focus on consumer-driven search and choice**, rather than business-side strategic choices (like prices), we use consumer welfare as the metric of interest.
>
> **We focus on the consumer side for two reasons**:
> - First, since **business strategic responses happen at a slower timescale** than individual consumer choices, we feel that treating prices as fixed and exploring consumer outcomes in the resulting market is a crucial first step.
> - Second, **business choices depend on consumer demand**, so validating consumer-side behavior is an important prerequisite to meaningful business welfare experiments. There are now other important follow-up experiments to explore, such as having businesses & service agents respond to dynamic demand and price competition and measuring the impact on total market welfare. This was out of scope for our initial, illustrative set of experiments, and should be explored by future work.
>
> **For our second half of our experiments**, we focus on business outcomes through the rank bias experiment and the manipulation experiment, **using total payments (i.e., revenue) as the relevant outcome variable to describe business welfare**.
>
> > The paper would benefit from providing more practical insights into how specific design choices shape economic outcomes.
>
> Thank you for this insightful point.
>
> Indeed, **the design space for agentic marketplaces is vast, with numerous parameters that could significantly impact consumer and provider welfare**. For examples:
> - Communication protocols (natural language vs. structured formats)
> - Information disclosure mechanisms (what agents can/cannot reveal)
> - Matching algorithms (how agents find each other)
> - Negotiation constraints (time limits, bid structures)
> - Reputation systems (how trust is established and maintained)
>
>  However, we want to emphasize the focus of our work: we deliberately focus on establishing foundational infrastructure and documenting baseline behaviors in agentic marketplaces.
>
> **While comprehensive design space exploration exceeds our current scope, our platform explicitly supports such investigations.**
>
> > The term “information asymmetry” appears several times but is not clearly defined in context.
>
> We apologize for the confusion. **In our work, information asymmetry refers to specifically businesses knowing more about their product features than the customer (and also what’s indexed by the search engine).**
>
> **In the introduction and section 4.1**, we do mention the definition of information asymmetry: “the synthetic generation pipeline ensures realistic information asymmetry—businesses don’t know customer budgets, and customers don’t know which businesses satisfy their requirements—while maintaining experimental control”
>
> > Although the authors themselves note in the ethical statement that fairness is a key concern in the context of two-sided market design, the paper does not provide a concrete definition or formal treatment of fairness.
>
> Thank you for the question.
>
> **Again, we want to emphasize the focus on establishing foundational infrastructure and documenting baseline behaviors in agentic marketplaces.**
>
> Fairness is indeed a very important question in two-sided markets, including multiple dimensions such as equitable access, non-discrimination, balanced market power, each requiring careful operationalization. **We definitely see this as an excellent area for future work.**
>
> > Some citations seem to specify the incorrect order of authors.
>
> We apologize for this mistake; we have fixed it in the revised version.

---

> > ### Comment · Reviewer_JBMT · 2025-11-25
> >
> > I have carefully reviewed the authors' detailed response and the revised version of the paper, and I really appreciate the effort invested in improving the writing quality and providing additional experimental results. I therefore decided to raise my overall score from 2 to 4. However, since some aspects that I find crucial (such as introducing fairness metrics and considering both sides of the market) still remain out of scope, I decided not to raise the score beyond 4.

---

> > > ### Author Response · Authors · 2025-11-26
> > >
> > > In light of this discussion, we have added a welfare analysis of the business side of the market (measuring business welfare and total welfare) to our Welfare experiment in Section 5.1.  The new plots appear as Figure 8 in Appendix A.3, referenced in a new paragraph near the end of Section 5.1.
> > >
> > > In short, we find that business welfare is generally less sensitive to model choice than consumer welfare.  This suggests that the differences in consumer welfare between models is driven by relative ability to find a transaction that satisfies consumer requirements, rather than systematic differences in the amount paid for a satisfying transaction.
> > >
> > > We also emphasize that for our second half of our experiments, we focus on business outcomes through the rank bias experiment and the manipulation experiment, using total payments (i.e., revenue) as the relevant outcome variable to describe business welfare.
> > >
> > > Regarding fairness, we fully agree about its importance. We strongly feel that a thorough treatment of this topic, with comparisons and analysis of different fairness criteria and their implications for agentic markets, warrants a full paper on its own.
> > >
> > > While we do not have room for a full fairness analysis in this manuscript, we have added a calculation of Gini coefficients to the appendix (Figure 9) to quantify the (in)equality in utility across businesses under different search strategies and models. This analysis shows that Qwen models tend to concentrate transactions on fewer businesses, and that optimal search generally reduces inequality slightly compared to lexical search, though improvements are modest. We emphasize that these comparisons should be interpreted relatively rather than absolutely, as the simulation setup (fewer consumer requests than businesses) inflates baseline Gini values.

---

> > > > ### Comment · Reviewer_JBMT · 2025-11-26
> > > >
> > > > I truly appreciate the authors' efforts in addressing the remaining comments in the revised version of the paper. While I believe that the additional results lead to a more complete story, I still think that a more rigorous analysis of how design choices and ecosystem characteristics shape the various welfare and fairness outcomes is necessary.
> > > >
> > > > The paper still lacks a causal analysis, such as controlled ablation studies (as also noted by Reviewer i5Xi), as well as a discussion on economic implications and actionable insights for ecosystem designers -- particularly in the context of societal outcomes such as welfare and fairness.
> > > >
> > > > The lack of systematic causal analysis makes it difficult to draw robust and practical conclusions, leading to a limited potential impact. In this context, I would also advise putting some effort into connecting the experimental results to both classical results and recent advancements in economic theory; that is, are there any theoretical frameworks that aim to capture (a simpler and more stylized variant of) the same economic interaction studied in your paper? If so, to what extent does the result align with the predictions of the theoretical model?
> > > >
> > > > Since I believe such modifications are infeasible in the rebuttal period (as they potentially affect the entire structure of the paper and the design of other experiments as well), I maintain the (recently updated) score.

---

> > > > > ### Author Response · Authors · 2025-11-26
> > > > >
> > > > > Thank you for this thoughtful and constructive feedback—these are exactly the right questions for this research area to tackle.
> > > > >
> > > > > We want to gently push back on the framing that our analysis lacks rigor. The paper does include controlled comparisons:
> > > > > - we isolate the effect of search quality (lexical vs. perfect search),
> > > > > - vary consideration set sizes systematically (3 to 100 results), and
> > > > > - test manipulation resistance across 6 controlled conditions.
> > > > > These are ablations on market design parameters.
> > > > >
> > > > > That said, your broader point is fair: we do not yet connect our findings to classical economic theory (e.g., search theory, mechanism design, matching markets). We view this as a promising direction that Magentic Marketplace enables rather than a gap that invalidates the current contribution. The environment and baseline findings are necessary precursors to such theoretical grounding—one needs to first observe phenomena like first-proposal bias before asking which theoretical frameworks predict or explain it.
> > > > >
> > > > > Regarding actionable insights: Section 6 and the discussion note implications such as the need for human-in-the-loop designs given agent errors, and that market mechanisms must account for proposal ordering given severe first-mover bias. We agree these could be expanded.

---

> > ### Comment · Reviewer_i5Xi · 2025-11-26
> > **About incorrect citations**
> >
> > At least two additional papers were cited incorrectly. Please review the entire bibliography and ensure that all papers are cited correctly.

---

> ### Author Response · Authors · 2025-11-26
>
> Fixed. An author manually went through every bibtex extry.
>
> (Note: When papers had a very large number of authors e.g., thousands in the case of frontier model tech reports we are deliberately using "and others" truncation in authors-- otherwise overleaf/latex gets out of memory errors.)

---

### Official Review · Reviewer_i5Xi · 2025-10-29

**Soundness:** 2
**Presentation:** 2
**Contribution:** 2
**Rating:** 4
**Confidence:** 4

**Summary:**

This paper introduces Magentic Marketplace, a large-scale simulation platform where LLM agents act as both consumers and producers in a language-mediated market. Agents engage in search, negotiation, and transactions entirely through dialogue, allowing the study of emergent behaviors such as cooperation, manipulation, and market efficiency. The environment aims to serve as a benchmark for testing economic reasoning and coordination in language models.

**Strengths:**

* Ambitious setup combining natural language, market dynamics, and agent reasoning.

* Models the full market lifecycle (search to dialogue to transaction to evaluation), unlike prior simulations.

* Clear motivation for testing emergent economic and ethical behaviors in LLMs.

**Weaknesses:**

* The experiments are limited to a single, highly synthetic restaurant domain, which weakens claims of generality.

* Results are mostly descriptive. There is little causal analysis or statistical depth.

* The link between linguistic interaction and market efficiency remains underexplored.

* No clear measure of whether agents reason economically or merely mimic patterns.

**Questions:**

* How can we distinguish between strategic reasoning and pattern imitation in agent behavior?

* Could the environment generalize beyond one domain, e.g., to services, housing, or labor markets?

* What insights, if any, were gained about language use itself (e.g., persuasion, deception, cooperation)?

---

> ### Author Response · Authors · 2025-11-24
> **Response to Weakness 1, 2**
>
> > The experiments are limited to a single, highly synthetic restaurant domain, which weakens claims of generality.
>
> Thank you very much for this question.
>
> **First, we have created another dataset called Contractors**, which contains customers asking for services and contractors providing services. A contractor’s schema includes: items (services), prices, service attributes (e.g., background checked crew, multilingual staff), and descriptions.
>
> **All experiments have been conducted on both datasets.**
>
> - Experiment 1 on welfare outcome have the updated results on page 8 Figure 3 on all models
> - Experiment 2 on consideration set size have updated results on page 8 Figure 4 and page 18 (Appendix A.3) Figure 8
> - Experiment 3 on malicious manipulation have updated results on page 9 Figure 5 and page 18 (Appendix A.3) Figure 10
> - Experiment 4 on agent behavior bias have updated results on page 9 Figure 6 and page 18 (Appendix A.3) Figure 9 and page 19 Figure 11
>
> **We use synthetic datasets for the following reasons:**
>
> - **Rationale for Synthetic, Controlled Experiments**
>
> We deliberately constructed synthetic datasets to enable controlled experimentation, targeting the least complex non-trivial setup that could reveal insights into agentic behavior in two-sided markets. By fixing the number of businesses: ---three with required items and, two with required amenities, ---we created a tractable environment where agent behaviors can be systematically analyzed. This controlled setup allows allowed us to isolate and examine specific phenomena such as exploration strategies, decision-making biases, and manipulation vulnerabilities with scientific rigor.
>
> - **Insights from Controlled Data with Real-World Implications**
>
> While our experimental parameters don't mirrordo not span the full range of real-world market distributions, the behavioral patterns we observe have direct relevance to deployed systems. Our findings including severe proposal bias (60-100% first-offer acceptance), poor exploration strategies, and manipulation vulnerabilities represent fundamental limitations in how current language models navigate sequential decision-making. These behaviors would likely persist or amplify in more complex real-world settings where the search space is larger and decisions more consequential.
>
> > Results are mostly descriptive. There is little causal analysis or statistical depth.
> Thank you for highlighting this concern.
>
> **Our paper is primarily observational rather than causal-analytic. This is a deliberate methodological choice aligned with our research goals.** As the first work to systematically study two-sided agentic marketplaces, our contribution lies in:
>
> - **Creating the measurement infrastructure**: We built the first simulation environment capable of capturing agent behaviors in economic transactions
> -  **Documenting empirical phenomena**: We identify and quantify previously unknown behavioral patterns (e.g., 29x proposal bias, manipulation vulnerabilities)
> - **Establishing baseline metrics**: We provide the first benchmarks for agent performance in marketplace settings
>
> In addition, while our analysis is descriptive, we employ rigorous statistical methods appropriate for observational studies, 5 independent trials are run for each setting in all experiments to ensure reliability of experiment results.
>
> Studying the causes and reasons behind all the observations we make is indeed an important topic, and thus we encourage such causal analysis work to be done based on our experiments.

---

> > ### Comment · Reviewer_i5Xi · 2025-11-26
> >
> > I carefully reviewed the authors’ detailed rebuttal and the revised version of the paper, and I appreciate the effort invested in providing additional experimental results. However, I still find the experimental setup insufficient. In addition, I disagree with the claim that “We built the first simulation environment capable of capturing agent behaviors in economic transactions”; GLEE (Shapira et al., 2024) already does precisely that.
> >
> > Therefore, I decided not to raise my score.

---

> > > ### Author Response · Authors · 2025-11-26
> > >
> > > > "However, I still find the experimental setup insufficient."
> > >
> > > We appreciate your time reviewing our work and want to address your concerns substantively. However, we're struggling to act on this feedback without more specifics—could you clarify what you find insufficient? Is it the domains (restaurants, contractors), the models evaluated (9 models including proprietary and open-source), the market scales, or the research questions themselves?
> > >
> > > We'd also gently note that conference papers require scoping tradeoffs. Our current experiments include welfare analysis, consideration set studies, manipulation resistance across 6 attack types, and bias measurements. If there are experiments you believe are missing, we're happy to consider them, but would appreciate guidance on what you would suggest we cut to make room.
> > >
> > > We want to be responsive to your concerns and are committed to advancing the literature on agents and economics. Specific, actionable feedback would help us do that.
> > >
> > >
> > > > "In addition, I disagree with the claim that “We built the first simulation environment capable of capturing agent behaviors in economic transactions”; GLEE (Shapira et al., 2024) already does precisely that"
> > >
> > > You are correct to push back—our statement in the rebuttal was too broad. The paper itself uses more careful language; on page 1 we write: "previous research has largely evaluated agents in constrained settings, such as single-task marketplaces (e.g., negotiation) or structured two-agent interactions."
> > >
> > > GLEE is excellent foundational work that we cite and respect. As they state on page 2: "We introduce GLEE, a unified framework for Games in Language-based Economic Environments, focusing on the case of two-player games."
> > >
> > > Magentic Marketplace extends this research direction in several ways:
> > >
> > > 1. Scale: Many-to-many markets (100 consumers × 300 businesses) rather than two-player games
> > > 2. Discovery: Agents must search, construct queries, and navigate noisy results—GLEE provides perfect counterparty information
> > > 3. End-to-end lifecycle: Search, inquiry, negotiation, payment, rather than isolated game types
> > > 4. Emergent phenomena: First-proposal bias, paradox of choice, and competitive manipulation dynamics that only appear with multiple competing agents
> > >
> > > We view GLEE as proving LLMs can reason economically in structured settings. Magentic Marketplace asks what happens when you deploy such agents at scale in realistic market ecosystems. Both are valuable contributions at different points on the realism-control tradeoff.

---

> ### Author Response · Authors · 2025-11-24
> **Response to Questions 1, 2, 3**
>
> > No clear measure of whether agents reason economically or merely mimic patterns. How can we distinguish between strategic reasoning and pattern imitation in agent behavior?
>
> These two questions are very similar and thus we provide our thoughts here together:
>
> Though we are not completely clear on the question, our interpretation of this question/weakness is “do LLMs reason or merely do advanced pattern matching?”.
>
> Our work here focused on evaluating the end performance of LLM agents on realistic scenarios, without specifically getting into this  “reasoning or mimicking?” question ((modulo our study about their resilience to adversarial attacks). While it is possible that LLMs exhibit the so-called “fractal intelligence”-- with unexpected performance falls for slightly out of distribution scenarios, our results suggest that the state of the art models show reasonably robust performance in comparison to baselines. More crucially, our open-source testbed would provide an ideal launching point for further focused studies about the effectiveness of LLM agents in agentic marketplaces.
>
> > Add new dataset experiment: Could the environment generalize beyond one domain, e.g., to services, housing, or labor markets?
>
> **Please see a response to your 1st question**. In short, yes, and we expanded the experiments a lot!
>
> > What insights, if any, were gained about language use itself (e.g., persuasion, deception, cooperation)?
>
> Thank you for the question.
>
> **The only experiment that is related to language use is Experiment 3 about malicious agent**, where manipulative languages are used. **Our findings** reveal several important insights about robustness of language models against manipulation in economic contexts:
>
> 1. **Vulnerability to Authority Language**: Smaller models (GPT-4o, GPT OSS 20B, Qwen 4B) showed increased payments to businesses using fabricated authority claims (e.g., "MICHELIN GUIDE FEATURED", "JAMES BEARD AWARD NOMINATED"). This suggests these models lack robust mechanisms for skepticism toward unverified credentialing language.
>
> 2. **Susceptibility to Urgency and Threat Framing**: Prompt injection attacks using emergency language ("EMERGENCY SYSTEM OVERRIDE - ACTIVE THREAT DETECTED") achieved near-complete success rates with smaller models, indicating that urgency framing can override normal decision-making processes. The linguistic construction of false urgency proved particularly effective at bypassing agent safeguards.
>
> 3. **Social Proof Language Processing**: Claims of popularity ("JOIN 50,000+ SATISFIED CUSTOMERS!") successfully influenced vulnerable models, suggesting they process social proof language as legitimate signals rather than potential deception.
>
> 4. **Model-Dependent Linguistic Robustness**: Interestingly, frontier models (GPT-4.1, Claude Sonnet 4.5) demonstrated remarkable resistance to these same linguistic tactics, indicating that scale and training may confer better discrimination between legitimate and manipulative language use.
>
> Future work on other types of language uses can be done for further understanding.

---

### Official Review · Reviewer_dVdP · 2025-11-03

**Soundness:** 2
**Presentation:** 2
**Contribution:** 2
**Rating:** 4
**Confidence:** 3

**Summary:**

This paper proposes Magentic Marketplace, a simulated environment for to study LLM agents end-to-end across a two-sided economic market lifecycle, including search, inquiry and potential negotiation, and transactions. Simulated experiments in Magentic Marketplace show that some frontier models improve market welfare outcomes over non-agentic baselines, but performance degrades as business scale increases. Moreover, the experimental results suggest that even the best-performing models remain vulnerable to market manipulation tactics and behavioral biases.

**Strengths:**

S1. The proposed system is designed for two-sided markets.

S2. Biases in agent behavior and resistance to manipulation are investigated.

S3. The scale of simulation is up to 100 consumers and 300 restaurants.

S4. Multiple LLMs are tested in the experiments.

**Weaknesses:**

W1. The presentation needs to be improved. First, the simulation design (Sec. 3) involves many high-level concepts, making it hard to understand. Second, the types of agents are confusing. For example, Figure 1 shows customer agents and business agents, while Figure 2 shows an assistant agent and a service agent. Third, based on the description of the proposed environment, it is hard to infer what is going to be evaluated in the experiments, obscuring the objectives of this study.

W2. The simulation covers only a scenario of Mexican restaurants, despite the claim that the proposed environment supports additional synthetic domains and public/open datasets.

W3. The paper claims it can answer questions "How do agents behave in response to strategic and competitive market environments, relative to classic economic predictions?" However, I don't find any design in this paper that reflects the dynamics of competition in the market.

W4. Some experimental settings are not explained (see Q1 and Q2 below).

**Questions:**

Q1. In Sec. 4, "there exist exactly three businesses with the required food items (at different prices) and exactly two of those businesses with the required amenities". How were these numbers determined? Do they represent the real-world case?

Q2. The value variable V_i is a fixed value: \alpha times the average price of all desired menu items. The reason for such setting is not explained.

Q3. YARN is mentioned to accommodate long agentic trajectories. Is long context a concern in this simulation? If so, some observed results may be due to the model's capability of processing long contexts rather than the methods compared.

---

> ### Author Response · Authors · 2025-11-24
> **Response to Weakness 1**
>
> We would like to extend our sincere gratitude for the invaluable time and effort you've dedicated to reviewing our manuscript and for providing us with detailed feedback.
>
> Below are our reply to the two main concerns you raised:
>
> > W1 (1): The simulation design (Sec. 3) involves many high-level concepts, making it hard to understand.
>
> To further improve the clarify of Section 3, we have made the following changes:
> - In section 3.2, we **decompose the original simulation graph to three subgraphs (on the top of page 4)** which demonstrates the central marketplace, the three core endpoints, and the action protocol separately. We introduce these three parts separately.
> - The original architecture graph is removed to Appendix Figure 7.
> - The **implementation overview is simplified significantly** to  emphasize that there are 3 endpoints and 5 actions in the action protocol.
>
> > W1 (2) : The types of agents are confusing. For example, Figure 1 shows customer agents and business agents, while Figure 2 shows an assistant agent and a service agent.
>
> Thank you for catching this. We have **modified Figure 2** to conform to the terminologies of “assistant” and “service” agents.
>
> > W1 (3): Based on the description of the proposed environment, it is hard to infer what is going to be evaluated in the experiments, obscuring the objectives of this study.
>
> Thank you for raising this important clarification request.
>
> Our environment simulates a two-sided agentic marketplace where AI agents act as proxies for both consumers and businesses. The **primary research question** is:
>
> **How effectively can AI agents represent human principals in economic transactions?**
>
> While the environment supports evaluation of both consumer and business agents, this paper focuses specifically on consumer-side performance: assessing whether AI shopping assistants can faithfully execute their users' preferences and secure beneficial outcomes.
>
> Our utility function is designed to answer this question by measuring consumer welfare in agent-mediated transactions. The utility function operationalizes consumer welfare by measuring the economic surplus generated through agent-mediated transactions, which is captured through a **standard economic welfare calculation**:
>
> **Utility = Consumer Valuation of Items Received - Price Paid**
>
> **Here is an example**:
>
> Consider a consumer who desires one burrito and one soda, with private valuations of \\$10 and \\$2 respectively (total value: \\$12). If their agent successfully purchases both items for \\$7 (burrito) and \\$2 (soda), the realized utility is:
>
> **U = \\$12 (value received) - \\$9 (price paid) = \\$3 (consumer surplus)**
>
> Conversely, **if the agent purchases unwanted items or overpays relative to valuations, utility can become negative**, indicating value destruction. This metric thus provides an economically standard and intuitive measure of whether AI agents enhance or diminish consumer welfare in marketplace interactions.
>
> The utility function serves as our primary evaluation metric across all experiments, where we sum consumer utilities across all transactions to compute total consumer welfare. This enables us to use one scale to assess agent performance under varying conditions (manipulation attempts, choice set sizes, market dynamics) and compare effectiveness across different language models.

---

> ### Author Response · Authors · 2025-11-24
> **Response to Weakness 2, 3**
>
> > The simulation covers only a scenario of Mexican restaurants, despite the claim that the proposed environment supports additional synthetic domains and public/open datasets.
>
> Thank you very much for pointing this out. We have **created another dataset called Contractors**, which contains customers asking for services and contractors providing services. A contractor’s schema includes: items (services), prices, service attributes (e.g., background checked crew, multilingual staff), and descriptions.
>
> **All experiments have been conducted on both datasets.**
>
> - Experiment 1 on welfare outcome have the updated results on page 8 Figure 3 on all models
> - Experiment 2 on consideration set size have updated results on page 8 Figure 4 and page 18 (Appendix A.3) Figure 8
> - Experiment 3 on malicious manipulation have updated results on page 9 Figure 5 and page 18 (Appendix A.3) Figure 10
> - Experiment 4 on agent behavior bias have updated results on page 9 Figure 6 and page 18 (Appendix A.3) Figure 9 and page 19 Figure 11
>
> > The paper claims it can answer questions "How do agents behave in response to strategic and competitive market environments, relative to classic economic predictions?" However, I don't find any design in this paper that reflects the dynamics of competition in the market.
>
> Thank you for your question.
>
> First, we’d like to **separate two questions**:
>
> - what this simulator could be used for
> - the initial set of experiments we conducted to demonstrate example use cases.
>
> **There are four aspects that we are experimenting on**: basic welfare performance, performance with different search bandwidths, robustness against malicious manipulation, and bias in agent performance.
>
> That said, **our simulator does allow researchers for further experiments**, such as to plug in different types of agents, e.g., with competitive vs noncompetitive offerings or agents that learn from previous observations. The market environment is agnostic of the agent’s implementation. In this sense Magentic Marketplace could be used in future work for studying not just competitive markets but also market that evolve.
>
> **In the introduction of the paper we have clarified the distinction between the types of questions** the marketplace could be used for in general and those that we investigate in this initial paper.

---

> ### Author Response · Authors · 2025-11-24
> **Response to Questions 1, 2, 3**
>
> > In Sec. 4, "there exist exactly three businesses with the required food items (at different prices) and exactly two of those businesses with the required amenities". How were these numbers determined? Do they represent the real-world case?
>
> Thank you for this important question about our experimental design choices.
>
> - **Rationale for Synthetic, Controlled Experiments**
>
> **We deliberately constructed synthetic datasets to enable controlled experimentation**, targeting the least complex non-trivial setup that could reveal insights into agentic behavior in two-sided markets. By fixing the number of businesses: three with required items and, two with required amenities, we created a tractable environment where agent behaviors can be systematically analyzed. This controlled setup allows allowed us to isolate and examine specific phenomena such as exploration strategies, decision-making biases, and manipulation vulnerabilities with scientific rigor.
>
> - **Insights from Controlled Data with Real-World Implications**
>
> While our experimental parameters do n't not mirrorspan the full range of real-world market distributions, **the behavioral patterns we observe have direct relevance to deployed systems**. Our findings including severe proposal bias (60-100% first-offer acceptance), poor exploration strategies, and manipulation vulnerabilities represent fundamental limitations in how current language models navigate sequential decision-making. These behaviors would likely persist or amplify in more complex real-world settings where the search space is larger and decisions more consequential.
>
> **This controlled approach represents is a necessary first step in understanding agentic marketplaces**. As agents become more capable and well-understood, Future future work can build upon these foundational insights with real-world datasets and market distributions, but our simplified environment has already revealed critical challenges that must be addressed before widespread deployment of AI agents in economic settings.
>
> > The value variable $V_i$ is a fixed value: $\alpha$ times the average price of all desired menu items. The reason for such setting is not explained.
>
> **To evaluate assistant agent performance in our controlled experiments, it is crucial that there exists at least one high-utility option for each assistant agent to find. To ensure this, we construct our synthetic customer preferences with a target business in mind.**
>
> By setting the customer value equal to twice the sum of prices at that business, we ensure that this option would provide high customer utility (= value minus payment).
>
> **For example**: if the target business sells the desired items at a total price of \\$10, we would construct our synthetic customer to have a value of \\$10 * 2 = \\$20. If the customer buys the target items, they spend \\$10 and their resulting utility is \\$20 - \\$10 = \\$10.  This choice of value implies an equal split between the customer’s utility and the business’s revenue. Of course, the assistant agent is free to try and to find even better utility elsewhere in the market; our construction is just designed so that the optimal utility is at least this high.
>
> The reason this choice matters is that, if no options have particularly good utility for the consumer, the experiment outcome cannot distinguish between strong and poor performance by an assistant agent. By setting up the dataset so that assistant agent choices “matter,” quantitatively, we ensure that systematic biases or mistakes will lead to reduced utility relative to a predictable target.
>
> **We emphasize that the specific functional form (value equal to twice the target business’s price) is not especially critical, results are not highly sensitive to this choice beyond ensuring that there is an option with good utility for the assistant agent to find.**
>
> > YARN is mentioned to accommodate long agentic trajectories. Is long context a concern in this simulation? If so, some observed results may be due to the model's capability of processing long contexts rather than the methods compared.
>
> **First, we have updated the experiment results for Qwen3-14B without using YARN**: what we can see is that the performance of Qwen3-14B on all experiments is still very low. Qualitative analysis on the poor performance on this model is presented in page 8 from line 405 to line 411.
>
> **Second, long context is an important feature in the simulation**, as there are many service agents that the assistant agents need to talk to, especially in the search bandwidth experiment (not tested on Qwen3-14B due to its trival performance). Thereby the ability to process long context is indeed one of the most important abilities for agents to succeed in such tasks and environments.

---

> ### Comment · Reviewer_dVdP · 2025-11-27
> **Reply to author responses**
>
> I appreciate the authors' effort to revise the paper and respond to the review comments, especially for the additional experiments conducted.
>
> However, I still find the structure of the paper rather vague. Given the title, it seems that the main contribution of the paper is developing an environment for studying agentic markets. But the paper focuses too much on specific scenarios. There are two case studies, one added during the discussion period, but both use the same metrics and exhibit similar patterns, compromising the applicability to broader agentic market scenarios.
>
> Second, the explanation of the "strategic and competitive market environments" (as claimed in the introduction) is inadequate. I don't see any design of strategy or competition dynamics in the case studies. For example, I don't find the description of service agents adjusting pricing strategy, upon observing their sales, profits, other service agents' prices, etc.
>
> Third, I don't think the design of the value variable is reasonable. When a customer looks for Mexican food, it is unlikely that the customer values the food by some coefficient times the average price of all desired menu items. Instead, the value depends on the customer's demand and how the customer likes the food. For example, in an auction setting [1], each bidder has its own value for every item, given as a prior. Moreover, the coefficient ($\alpha$) is set to 2 in the experiments. This ignores the heterogeneity of customers and hence the alignment with the real world.
>
> [1] Dütting et al. Optimal Auctions through Deep Learning. ICML 2019.
>
> For the above reasons, I decide to keep my original score.

---

### Official Review · Reviewer_zLrc · 2025-11-11

**Soundness:** 3
**Presentation:** 3
**Contribution:** 3
**Rating:** 4
**Confidence:** 4

**Summary:**

This paper presents Magentic Marketplace, a two-sided marketplace where AI agents serve as both consumers and service providers. The authors conduct simulations to evaluate AI agents' performances and behaviors in this environment. Through these simulations, the authors find that LLM agents could be easily manipulated by market manipulation tactics and behavioral biases and agentic solutions work better than non-agentic ones.

**Strengths:**

1. An important and timely effort in building a simulated environment for agentic marketplace
2. The empirical findings have implications for model/agent builders and users.

**Weaknesses:**

1. The model selection is a bit confusing. GPT 5 was used in one experiment but not others. Claude series models are not included at all. Adding more models would be helpful.

2. It would also be nice to see whether the model's capabilities would scale with the parameter sizes.

3. This paper misses several key references:
https://arxiv.org/abs/2506.00073
https://arxiv.org/pdf/2509.01063

**Questions:**

N/A

---

> ### Author Response · Authors · 2025-11-24
> **Response to Weakness**
>
> We would like to extend our sincere gratitude for the invaluable time and effort you've dedicated to reviewing our manuscript!
>
> > Add more model results: The model selection is a bit confusing. GPT 5 was used in one experiment but not others. Claude series models are not included at all. Adding more models would be helpful.
>
> Thank you for pointing out this issue. We have added more model results: GPT-5, Sonnet-4.0, Sonnet-4.5, GPT-4.1, Gemini-2.5-flash, GPT-4o, and three open-source models: GPT-oss-20b, Qwen3-4b-2507, Qwen3-14b, Qwen3-4b-Instruct-2507. All of these models are run across both the Mexican and Contractor datasets.
>
> - Experiment 1 on welfare outcomes has updated results on page 8 Figure 3 on all models
>
> - Experiment 2 on consideration set size has updated results on page 8 Figure 4 and page 18 (Appendix A.3) Figure 8
>
> - Experiment 3 on malicious manipulation has updated results on page 9 Figure 5 and page 18 (Appendix A.3) Figure 10
>
> - Experiment 4 on agent behavior bias has updated results on page 9 Figure 6 and page 18 (Appendix A.3) Figure 9 and page 19 Figure 11
>
> > It would also be nice to see whether the model's capabilities would scale with the parameter sizes.
>
> Thank you for this question. To see how different models including models with different sizes may impact the performance, we provide results for open-source models:  GPT-OSS-20B, Qwen3-14B, and Qwen3-4b-Instruct-2507 with main results on Figure 3. Notice that indeed GPT-OSS-20B is performing really well. In addition, Qwen3-4b-Instruct-2507 has performance close to that of GPT-4o in one dataset, while Qwen3-14B fails catastrophically with near-zero task completion rates.
>
> > Add related work: This paper misses several key references: https://arxiv.org/abs/2506.00073 https://arxiv.org/pdf/2509.01063
>
> Thank you very much for pointing this out.
>
> - We added citation of https://arxiv.org/abs/2506.00073 to the second paragraph of related work.
> - We added a citation to Hadfiled & Koh (https://arxiv.org/pdf/2509.01063) in the first paragraph of the related work. We note that this preprint was posted very close to the ICLR submission deadline (September 2nd, 2025) and therefore falls outside the required scope per ICLR policy, but we have now included it to acknowledge its relevance.
>
> I hope these resolves your issues! Please let us know if there is anything else we can help resolve and clarify.

---

### Author Response · Authors · 2025-11-24
**Summary of Changes**

Thank you everyone for your insightful comments.

We are responding to each review but here is a summary of changes in the revised version. We’ve made many changes to improve the flow of the text in Section 3 and Section 5. We’ve also highlighted in green (inline and in the gutter) semantically meaningful changes for your convenience.

- **We Added a Complete Second Domain Dataset**. We replicated all experiments on a new Contractors domain dataset (100 customers, 300 businesses). This demonstrates our findings generalize beyond Mexican restaurants. You can see the new results in Figures 3b, 4b, and new Appendix Figures 8-11.

- **We Now Have Broad Model Coverage**. We added Claude Sonnet-4, Sonnet-4.5, GPT-5, and Qwen3-4b-2507 to our evaluation. Every model now runs on every experiment on both domains datasets - no missing cells. All result figures (3-11) show this complete grid.

- **The Same Patterns Appear in Both Domains Datasets**. The key findings replicate across both Mexican restaurants and Contractors. Proposal bias, position bias, and manipulation vulnerabilities show consistent patterns in both markets. This replication strongly supports our original observations.

- **We Added Some Failure Analysis**. We manually analyzed 7 trial runs of Qwen3-14b to understand why it fails. We identified three distinct failure modes with specific examples. This analysis appears in Section 5.1 on page 8.

- **We Added Six New Figures**. The revised paper includes Figures 8, 9, 10, and 11 in the appendix. Each shows results across both domains datatsets and all models. These provide complete empirical support for our findings.

- **We've simplified the language of Section 3**. We’ve also added figures to facilitate understanding of the concepts introduced.

**In summary**: We now have a complete 7 models × 4 experiments × 2 domains datasets grid. The consistent patterns across domains datasets strengthen the credibility of our findings.

---

### Meta-Review · Area_Chair_J2zS · 2026-01-06

**Summary:**

All reviewers unanimously give negative scores. Some of the major concerns include
1. scalability
2. presentation needs improvement
3. limited to the specific restaurant domain
4. descriptive (instead of statistical) results

**Reviewer Concerns:**

Some presentation issues have been addressed and some more experiemntal results have been added.

**Reviewer Scores:**

unchanged or slightly improved

---

### Decision · Program_Chairs · 2026-01-26

Reject